# The PACT Network: PRL, ARL, CNNM, and TRPM Proteins in Magnesium Transport and Disease

**DOI:** 10.3390/ijms26041528

**Published:** 2025-02-12

**Authors:** Jeffery T. Jolly, Jessica S. Blackburn

**Affiliations:** 1Department of Molecular and Cellular Biochemistry, University of Kentucky, Lexington, KY 40536, USA; 2Markey Comprehensive Cancer Center, University of Kentucky, Lexington, KY 40536, USA

**Keywords:** metal homeostasis, ion transport, CBS-domain, phosphatase of regenerating liver, PTP4A3, channelopathies, cancer, therapeutic development

## Abstract

Magnesium, the most abundant divalent metal within the cell, is essential for physiological function and critical in cellular signaling. To maintain cellular homeostasis, intracellular magnesium levels are tightly regulated, as dysregulation is linked to numerous diseases, including cancer, diabetes, cardiovascular disorders, and neurological conditions. Over the past two decades, extensive research on magnesium-regulating proteins has provided valuable insight into their pathogenic and therapeutic potential. This review explores an emerging mechanism of magnesium homeostasis involving proteins in the PRL (phosphatase of regenerating liver), ARL (ADP ribosylation factor-like GTPase family), CNNM (cyclin and cystathionine β-synthase domain magnesium transport mediator), and TRPM (transient receptor potential melastatin) families, collectively termed herein as the PACT network. While each PACT protein has been studied within its individual signaling and disease contexts, their interactions suggest a broader regulatory network with therapeutic potential. This review consolidates the current knowledge on the PACT proteins’ structure, function, and interactions and identifies research gaps to encourage future investigation. As the field of magnesium homeostasis continues to advance, understanding PACT protein interactions offers new opportunities for basic research and therapeutic development targeting magnesium-related disorders.

## 1. Introduction

Magnesium is an abundant intracellular metal often overlooked as a mediator of cellular signaling due to its tight regulation. Historically, intracellular magnesium concentrations were thought to remain nearly constant due to magnesium’s slow turnover across the plasma membrane in the absence of stimuli and the limited reports of significant fluctuations [1]. However, advances in detection technologies and methodologies revealed subtle changes in intracellular magnesium levels, which have since been shown to mediate a wide range of cellular signaling events [2,3,4]. Given its vast physiological importance, disruptions in magnesium homeostasis have been implicated in numerous human diseases, including neurological, cardiovascular, and endocrine disorders [5,6,7]. Beyond these “canonical” disease associations, magnesium dysregulation is increasingly recognized as a contributor to oncogenesis and cancer signaling [8,9,10,11,12].

This review explores the emerging interactions between proteins in the PRL (phosphatase of regenerating liver), ARL (ADP ribosylation factor-like GTPase family), CNNM (cyclin and cystathionine β-synthase domain magnesium transport mediator), and TRPM (transient receptor potential melastatin) families, collectively referred to here as the PACT network. These proteins have well-documented individual associations with various diseases, including cancer. Here, we summarize the structural basis of their interactions, their regulatory mechanisms, and the downstream signaling consequences of these functions. Furthermore, as each member of the PACT network represents a promising drug target, disrupting their co-interactions may offer novel therapeutic opportunities, and these possibilities are highlighted throughout.

While substantial physical evidence supports the existence of the PACT network, the physiological relevance and mechanistic details of these interactions remain an area of active research. This review also highlights current gaps in the field and aims to encourage further investigation. The emergence of the PACT network is an exciting addition to the field of magnesium transport research. Understanding how these proteins modulate one another could provide critical insights into the pathophysiology of magnesium dysregulation and the broader mechanisms underlying cellular magnesium homeostasis.

## 2. Cellular Magnesium Homeostasis

### 2.1. Magnesium and the Cell

Divalent metals mediate a wide range of cellular signaling pathways through mechanisms such as acting as enzymatic cofactors or inhibitors, interacting with proteins to alter their structure and function, serving as secondary messengers in signal transduction, and regulating membrane potentials and ion channel activity. Among these, magnesium is the most abundant divalent metal and the second most abundant intracellular metal, surpassed only by potassium. Magnesium concentrations vary across tissues, with total serum magnesium levels in healthy adults typically ranging from 0.85 to 0.95 Mm [13]. Bone contains the highest magnesium concentration (43.2 mM), followed by muscle, soft tissue, and red blood cells at 9, 8.5, and 2.5 mM, respectively [14,15,16,17,18].

In contrast to serum concentrations, intracellular magnesium concentrations are relatively high, with reported ranges between 5–20 mM [19] and 10–30 mM [20], averaging 17–20 mM [21,22,23]. However, this intracellular magnesium is not evenly distributed. Most intracellular magnesium is bound to proteins and phosphorylated metabolites, particularly ATP, approximately ~90% of which is magnesium-bound due to its high abundance (3–10 mM) and strong affinity for magnesium (K_d_ ~0.1–1 mM) [1,24,25,26,27]. This Mg-ATP complex neutralizes the negative charge of ATP’s polyphosphate group, stabilizes its conformation, and creates additional enzyme interaction sites, thereby enhancing binding energy and forming a biologically active complex [28]. Only a small proportion (1–5%, or 0.5–1.2 mM) of intracellular magnesium is freely available in its ionized form [5,19,20,29]. This small pool of free magnesium plays a critical regulatory role in cellular processes and is tightly controlled by the cell.

Magnesium is a cofactor for over 300 different enzymatic reactions within the cell [29]. Fluctuations in magnesium concentrations regulate key enzymatic pathways. For instance, many glycolytic enzymes are magnesium-dependent, and glucose metabolism is strongly influenced by intracellular magnesium levels [30,31]. Magnesium is also essential for the activity of enzymes such as Na^+^/K^+^-ATPase, which modulates membrane potential and osmolarity, and adenylate cyclase, which regulates cAMP production [31,32]. Intracellular magnesium levels further impact diverse signaling pathways, including protein synthesis, DNA replication, and oxidative metabolism, as extensively reviewed elsewhere [3,33,34,35,36,37,38]. Due to its broad influence, magnesium homeostasis is maintained by a sophisticated network of proteins that tightly regulate its transport.

### 2.2. Established and Emerging Mechanisms of Magnesium Homeostasis

The primary proteins involved in magnesium homeostasis in eukaryotes include “mitochondrial RNA splicing 2” (Mrs2) [39], “solute carrier family 41” (SLC41) [40], “membrane Mg^2+^ transporter 1” (MagT1) [41], “non-imprinted in Prader–Willi/Angelman syndrome protein” (NIPA) [42], “membrane Mg^2+^ transporters” (MMgTs) [43], “huntingtin-interacting protein 14 transporters” (HIP14) [44], “cyclin and cystathionine β-synthase domain magnesium transport mediators” (CNNMs) [45], and “transient receptor potential melastatin 6/7” (TRPM6/7) [46]. These proteins have varying ion specificities and subcellular localizations. HIP14, MRS2, and MMgT1/2 primarily mediate intracellular magnesium transport, while MagT1, NIPA, SLC141, CNNM, and TRPM6/7 proteins facilitate magnesium transport across the plasma membrane [7].

Although these transporters have been studied extensively, their signaling and regulatory mechanisms remain incompletely understood [47,48,49]. Recent research has identified two key regulators of CNNM and TRPM6/7 activity: the phosphatase of regenerating liver (PRL) family and the ADP ribosylation factor-like GTPase 15 (ARL15) protein. These proteins likely interact in a PRL/ARL/CNNM/TRPM signaling network, which we term here as the PACT network for simplicity. A structural depiction of these proteins is shown in Figure 1.

### 2.3. Disruption of Magnesium Homeostasis and Cancer

Magnesium homeostasis in the body is primarily maintained through passive absorption in the small intestine and urinary excretion via the kidneys [29]. Mediated by proteins such as CNNMs and TRPM6/7, transport pathways can be upregulated or downregulated to sustain magnesium balance [54]. Disruption of this homeostasis can manifest in various diseases, including hypomagnesemia, often arising from dietary deficits and dysregulated magnesium transport in the intestines and kidneys [55]. In addition to its role in cardiovascular, neurological, and endocrine disorders [7,56,57,58,59,60,61], magnesium dysregulation has been implicated in oncogenesis and cancer progression.

The relationship between dietary magnesium and cancer is complex and has been studied since 1935 [62]. Despite over 80 years of research, no consensus exists on whether dietary magnesium promotes or inhibits cancer, as studies report conflicting findings [10,11]. This inconsistency may arise from magnesium’s regulation of overlapping physiological processes, such as inflammation and immune activation, which influence cancer progression in opposing ways [63,64,65,66,67].

At the cellular and molecular levels, the role of magnesium in oncogenic signaling is more apparent. Increased cellular magnesium levels or aberrant expression of magnesium transporters enhances several oncogenic phenotypes, including proliferation, apoptosis resistance, metastatic potential, and metabolic reprogramming to favor glycolysis [9,68]. Dysregulated magnesium homeostasis has been proposed as a biomarker of cancer progression. For example, advanced imaging techniques like X-ray fluorescence microscopy (XFRM) have detected increased magnesium concentrations in colon cancer lesions compared to adjacent non-tumor tissue [69]. Given the involvement of the PACT proteins in cancer, discussed below, understanding how these proteins regulate magnesium homeostasis could offer valuable insights into oncogenic mechanisms and therapeutic targets.

## 3. Individual PACT Proteins

### 3.1. Phosphatase of Regenerating Liver (PRL)

The phosphatase of regenerating liver family includes PRL-1, PRL-2, and PRL-3, which share a conserved structure featuring a central five-stranded beta-sheet surrounded by six alpha-helices and an unstructured C-terminus [52,70,71]. These PRL proteins are believed to function interchangeably within the PACT network, as evidenced by their high sequence identity (76–87% conserved) [72]. Each PRL protein contains a single dual-specificity phosphatase domain and a C-terminal prenylation (CaaX) motif that localizes them predominantly to the plasma membrane—both features are essential for their oncogenic functions [72,73,74,75,76,77].

PRLs, particularly PRL-3, are implicated in cancer metastasis and serve as negative prognostic markers, with high expression levels correlating with increased metastatic potential and poor patient outcomes across many cancer types [78,79,80,81,82,83,84,85,86,87,88,89,90,91,92,93,94]. Beyond cancer, PRLs play essential roles in development. In *Drosophila*, PRLs are crucial for establishing cellular polarity and promoting cell survival during early retinal development [95]. During embryogenesis, tissue-specific expression patterns of the PRLs in fruit flies, lancelets, and zebrafish suggest conserved roles in central nervous system development [96]. Additionally, PRL-1 and PRL-2 are implicated in spermatogenesis, and a double knockout of PRL-1/PRL-2 in mice is embryonically lethal. Interestingly, the individual knockouts of PRL-1 or PRL-2 produce viable offspring, indicating redundancy in their functions [97]. PRL-3 knockout mice are viable but exhibit mild developmental defects, including reduced body size in males and impaired glucose metabolism [98,99].

PRL-3 is expressed in developing blood vessels and fetal heart tissue but is absent in their mature counterparts, highlighting its therapeutic potential in cancer [100]. PRLs also promote angiogenesis by modulating VEGF-related pathways, which has broad implications in developmental and cancer biology [101,102,103,104,105]. Collectively, these findings suggest that PRLs are essential for normal development but can drive oncogenesis and metastasis when dysregulated.

While PRLs have demonstrated phosphatase activity [106,107,108,109,110,111,112,113,114], their primary substrates remain unclear. They dephosphorylate a broad range of protein substrates, including ezrin, keratin 8 (KRT8), p38 MAP kinase (MAPK), and phosphatase and tensin homolog (PTEN) [115,116,117,118,119], as well as non-protein substrates like phospho-nucleotides and phosphatidylinositol phosphates (PIPs) [120,121,122,123]. Given their plasma membrane localization, PIP lipids are strong candidates as the primary biological substrates for PRL phosphatase activity. This broad substrate range has placed PRLs in numerous oncogenic signaling networks [124,125,126]. However, as discussed below, their emerging role as CNNM-binding proteins adds complexity to their mechanistic characterization [127,128].

### 3.2. ADP-Ribosylation Factor-like GTPase 15 (ARL15)

The ADP-ribosylation factor-like (ARL) GTPases are members of the Ras superfamily, forming an evolutionary branch of the ADP-ribosylation factor (ARF) GTPase family [129]. Among the twenty ARL proteins identified [130], ARL15 has emerged as a critical component of the PACT network. Like other ARL proteins, ARL15 adopts a Ras-like structure comprising a central six-stranded beta-sheet surrounded by five alpha-helices [131]. Despite their conserved structure and ubiquitous expression, ARL proteins exhibit diverse functions, localizations, and activities, many of which remain poorly characterized [132,133].

ARL15, in particular, is relatively understudied, with limited information on its enzymatic activity, regulation, and localization. However, it has been implicated in various human diseases, including cardiovascular disease, diabetes, obesity, and rheumatoid arthritis [134,135,136,137]. Its role in cancer, while less explored, is intriguing. ARL15 was first linked to cancer as an independent prognostic factor in cutaneous melanoma (CM), a particularly deadly skin cancer. High ARL15 expression was associated with better patient outcomes, whereas low expression correlated with poor prognosis [138]. Conversely, high ARL15 expression in colon cancer was linked to the upregulation of metabolic and lipogenic proteins, including FASN, AKT, SREBP-1 (p125), and AMPK, promoting migration and invasion in cancer cell models [139].

Further studies have revealed a role for ARL15 in TGFβ signaling, where it interacts directly with Smad4 to enhance Smad complex assembly. ARL15 mutations, such as A86L and T64N, have been used to delineate its active GTP-bound and GDP-bound inactive states, which influence its regulatory role in TGFβ signaling [140]. Notably, the TGFβ signaling pathway can be tumor-suppressive or pro-metastatic, depending on the context, adding complexity to ARL15’s role in cancer [141]. These findings highlight the need for further investigation into ARL15’s mechanisms and implications in human disease.

### 3.3. CBS-Pair Domain Divalent Metal Cation Transport Mediators (CNNMs)

The CBS-pair domain divalent metal cation transport mediator (CNNM) family comprises four integral membrane proteins (CNNM1-4), initially identified as “ancient conserved domain proteins” (ACDPs) due to their resemblance to cyclins. These proteins were later renamed CNNMs to reflect the cystathionine β-synthase (CBS) domain pair they contain [142,143]. CNNMs are large, multi-domain proteins that include an extracellular N-terminal region, a DUF21 (domain of unknown function) transmembrane domain, a Bateman module, and a C-terminal cyclic nucleotide monophosphate-binding homology (CNBH) domain. The Bateman module, containing two CBS domains, has approximately 90% sequence identity among CNNMs [45,144] and is essential for magnesium transport and regulation [143]. CNNMs are highly evolutionarily conserved and have likely evolved from their orthologous MgtE and CorB/C predecessors found in prokaryotes. However, prokaryotic orthologs for TRM6/7 and the PRLs have yet to be identified, suggesting the CNNMs might be the oldest members of the PACT network, with perhaps the other members arising from further evolutionary complexity [45,49].

CNNM2 deletion is embryonically lethal in mice, and the heterozygous CNNM2 or CNNM4-deficient animals have lower magnesium levels than their wild-type counterparts [145]. The primary function of CNNM proteins is in magnesium homeostasis, although whether they act as direct magnesium transporters or modulators of other transport mechanisms is debated [146,147,148,149,150]. CNNM4, for instance, facilitates magnesium extrusion via Na^+^/Mg^2+^ exchange and is essential for intestinal magnesium absorption, as evidenced by hypomagnesemia in CNNM4 knockout mice [147]. CNNM2 knockouts similarly exhibit severe hypomagnesemia and are embryonic lethal, while CNNM2 heterozygous mice develop hypomagnesemia with increased serum calcium levels [151]. In cell cultures, CNNMs facilitate magnesium efflux, particularly following exposure to high magnesium concentrations (40 mM) [152,153]. Despite the similarities between these proteins, functional differences exist among CNNM family members. CNNM2 and CNNM4 exhibit the highest magnesium efflux capacities, CNNM1 shows moderate activity, and CNNM3 demonstrates minimal to no efflux capability [154].

In addition to mediating magnesium efflux, CNNMs also contribute to magnesium influx. Under hyperpolarization conditions, when the inside of the cell membrane becomes negatively charged, overexpression of both CNNM2 and CNNM3 has been shown to increase the cellular current in HEK293 cells. This effect was further enhanced by magnesium supplementation in the assay media, suggesting that CNNMs facilitate magnesium import under these conditions [155,156]. Radiolabeled magnesium experiments confirmed that CNNM2 overexpression increased magnesium uptake in HEK293 cells, a process dependent on sodium exchange. Notably, CNNM2-mediated magnesium uptake was abolished by treatment with 2-APB, a TRPM7 inhibitor [148,157].

In contrast, the knockdown of CNNM4, a homolog with the highest magnesium export capacity, increased intracellular magnesium levels in cell culture and enhanced mouse colon cancer malignancy. Polyps of CNNM4-deficient mice exhibited higher magnesium levels than their wild-type counterparts [153,158]. However, it should be noted that the reduction of CNNM4 levels does not drive oncogenesis alone, as evidenced by the tissue-specific silencing of CNNM4 in the livers of mice [159,160]. Furthermore, the silencing of CNNM4 in murine hepatocytes results in an accumulation of magnesium, as evidenced by an increased fluorescent signal produced by MagS and MagS-TP probes, suggesting the loss of CNNM4 results in the loss of cellular magnesium homeostasis [161]. Collectively, these findings indicate that CNNM proteins play multifunctional roles in magnesium regulation [148].

Mutations and misexpression of CNNMs are associated with various developmental defects. Pathogenic mutations in CNNM2 have been linked to hypomagnesemia, intellectual disabilities, and epilepsy [162,163,164]. Knockdown of CNNM2 in zebrafish disrupted several developmental processes, resulting in enlarged pericardial cavities and notochord defects. These phenotypes were attributed to reduced total magnesium content in the developing fish and were partially rescued by co-injection of wild-type CNNM2 mRNA [157]. Mutations in CNNM4 can cause Jalili Syndrome, a rare autosomal recessive condition characterized by cone–rod dystrophy of the retina and amelogenesis imperfecta [147,165,166,167,168]. Additionally, mutations in CNNM1 and CNNM3 CBS-pair and CNBH domains have been linked to schizophrenia, though the underlying mechanisms remain unclear [169].

CNNMs may also play roles in cancer. For example, CNNM4 has been proposed as a prognostic marker in colon cancer, where its mRNA levels are frequently reduced in metastatic disease [153,170]. The magnesium efflux function of CNNM4 has tumor-suppressive properties, and CNNM4 knockout enhanced malignant progression in a spontaneous colon adenocarcinoma mouse model (Apc^Δ14/+^) [153]. In contrast, CNNM3, a homolog with minimal magnesium export capacity, has been described as an oncogene, as its overexpression enhanced tumor growth in a DB-7 mammary cancer mouse xenograft model [155]. Taken together, these findings highlight the critical and multifunctional roles of CNNMs in human disease [45,171].

### 3.4. Transient Receptor Potential Melastatin (TRPM)

The transient receptor potential melastatin (TRPM) protein family is a subgroup of the transient receptor potential superfamily, consisting of eight ion channel proteins (TRPM1-8). Based on structural homology and ion permeability, the TRPM family is further divided into four subgroups: TRPM1/3, TRPM2/8, TRPM4/5, and TRPM6/7 [172,173,174]. Among these, TRPM7 is the only TRPM protein currently identified as part of the PACT network, with TRPM6, which shares 57% sequence homology, predicted to have similar functions [175]. TRPM6 and TRPM7 are large proteins (2220 and 1865 amino acids, respectively) that likely form tetrameric structures to create a central ion-conducting pore. They can assemble into both homomeric and heteromeric channels, though the extent of this heteromerization remains unclear [176,177]. These heteromeric combinations of TRPM proteins can alter channel properties, including activity, pH sensitivity, and ion preference, compared to their homomeric forms [178,179,180].

TRPM6/7 proteins contain three structural domains: an N-terminal melastatin homology region (MHR), a transmembrane domain, and a C-terminal kinase domain. The conserved TRP helix and coiled-coil segment within the transmembrane domain are critical for plasma membrane anchoring and tetrameric assembly [176,177,181,182].

TRPM proteins function as biological sensors for external stimuli, including temperature, pain, vision, taste, and mechanical shear. They act as ion channels with diverse gating and ion specificity properties [181]. TRPM6 and TRPM7 are primarily involved in magnesium ion uptake and reabsorption, particularly in the intestinal and renal systems [183,184]. These channels are also permeable to calcium and other divalent metals such as zinc [183,185,186,187,188]. Collectively, TRPM6 and TRPM7 perform similar but non-redundant functions, with their primary role being the regulation of divalent metal uptake.

The developmental roles and disease implications of TRPM6 and TRPM7 are substantial. Homozygous deletion of either TRPM6 or TRPM7 is embryonic lethal in mice [189,190]. TRPM6 heterozygous deletions are viable but result in hypomagnesemia [190]. In adult mice, inactivation of TRPM6 leads to shortened lifespans and metabolic defects, which can be mitigated through magnesium supplementation [191]. In humans, mutations of the TRPM6/7 proteins are linked to hypomagnesemia and other conditions, including macrothrombocytopenia and trigeminal neuralgia [192,193,194,195,196].

TRPM6 and TRPM7 also have emerging roles in cancer metastasis and oncogenesis. High TRPM7 expression correlates with an increased risk of invasive and metastatic disease in many cancer types, including breast, pancreatic, bladder, gastric, ovarian, and colorectal cancer [197,198,199,200,201]. In cell culture, knockdown of TRPM7 has been shown to impair various oncogenic signaling pathways. For example, Meng et al. (2013) demonstrated that TRPM7 knockdown in MDA-MB-435 cancer cells reduced their invasion and migration capacity and decreased phosphorylation of SRC and MAPK [197]. Similarly, Chen et al. (2017) reported that TRPM7 knockdown in pancreatic cancer cells reversed the epithelial-mesenchymal transition (EMT), a key process in cancer progression, by downregulating matrix metalloproteases and increasing E-cadherin expression [202]. Similar findings have been reported in colon cancer cells [201]. Conversely, the overexpression of TRPM7 promotes cell migration, invasion, and proliferation in bladder and lung cancer models [203,204,205]. Ultimately, increased TRPM7 expression significantly contributes to various hallmarks of cancer [174].

TRPM6 has been studied less extensively in cancer than TRPM7. Pugliese et al. (2020) found that while TRPM6 was upregulated in colorectal cancer samples, there was no clear correlation between TRPM6 levels and tumor grade or stage [206]. However, Zhang et al. (2014) reported that TRPM6 knockdown inhibited magnesium uptake and cell proliferation in neuroblastoma cells, suggesting that TRPM6 may share or complement TRPM7 functions [207].

It is important to note that while most studies suggest TRPM6/7 act as potential oncogenes, the cumulative clinical significance of their expression is complex. A comprehensive analysis of TRPM channel expression in cancer by Qin et al. (2020) revealed that TRPM6 and TRPM7 transcript levels vary significantly, with upregulation and downregulation reported depending on the cancer type [208]. Current research indicates that TRPM6/7 channels have important roles in regulating cellular metal concentrations and are implicated in human disease, though their exact roles remain to be fully defined.

## 4. Molecular Basis of the PACT Network

### 4.1. The First Connection: PRL and CNNM Interactions

The connection between PRL proteins and CNNMs was first established in 2014 when PRLs were found to bind to CNNMs and modulate their magnesium transport activities [153,209]. This interaction is evolutionarily conserved across all PRL and CNNM homologs, involving structural elements of the PRL phosphatase domain and the conserved loop of the CNNM Bateman module [210]. This interaction is independent of the catalytic activity of PRLs, as mutations that disable phosphatase activity, such as PRL-3 (C104D), do not disrupt CNNM binding. Other point mutants, such as PRL-3 (R138E) and CNNM3 (D426A), disrupt this interaction while preserving protein activity, providing valuable tools for mechanistic studies [52,152,153,155,209,211]. Multiple crystal structures of PRL–CNNM Bateman module complexes are available in the Protein Data Bank (e.g., 5K22, 5MMZ, 5LXQ, 5TSR, 5K23, 5K24, and 5K25), and PRL–CNNM interactions have been confirmed through co-immunoprecipitation studies in various cell lines [52,152,211].

Initial studies suggested that PRLs inhibit CNNM-mediated magnesium efflux. Using the fluorescent magnesium indicator Magnesium Green, studies by Funato et al. (2014) and Gulerez et al. (2016) demonstrated that the overexpression of CNNM4 in HEK293 cells caused magnesium efflux (quantified by a ~50% decrease in indicator intensity) when the culture medium was switched from high to low magnesium conditions. This effect was blocked by co-overexpression of PRL-3, implicating PRLs in the retention of intracellular magnesium [152,153]. Similarly, Hirata et al. (2014) showed that the deletion of the CBS domains of CNNM2/4 prevented magnesium export in a similar experimental setup [154]. However, whether or not this export activity is physiologically significant remains unclear. These findings initially supported the hypothesis that PRLs increase intracellular magnesium by inhibiting CNNM-mediated export.

Subsequent studies have proposed an additional mechanism in which PRLs enhance magnesium influx through CNNMs. Evidence for this hypothesis came from experiments in which CNNM3 was overexpressed in HeLa cells. Using Mag-fura2 (another fluorescent magnesium indicator), researchers quantified an increase in magnesium influx upon adding 10 mM MgCl_2_ to previously magnesium-free culture media. A CNNM3 mutant deficient in PRL-3 binding (G433D) showed a slight but significant reduction in magnesium influx compared to the wild-type CNNM3, suggesting a PRL-dependent role in magnesium uptake [209]. These results raised questions about the dual roles of CNNMs in mediating both magnesium efflux and influx, depending on PRL interactions.

The PRL–CNNM interaction has significant implications for cancer biology. In DB-7 mouse mammary tumor cells, overexpression of CNNM3 enhanced anchorage-independent survival in soft agar colony-forming assays. However, overexpression of CNNM3 (D426A), a mutant deficient in PRL binding, did not enhance anchorage-independent growth, indicating the effect was PRL-dependent. Similarly, while wild-type CNNM3 overexpression did not impact cell proliferation regardless of magnesium availability, CNNM3 (D426A)-expressing cells ceased to proliferate under magnesium-depleted conditions, suggesting that the PRL–CNNM interaction is crucial for maintaining magnesium homeostasis in such contexts [155]. Parallel studies with PRL-3 mutants support this conclusion. For instance, in a B16 mouse melanoma lung colonization model, wild-type PRL-3 enhanced tumor colonization, while the PRL-3 (C104S) mutant, deficient in both phosphatase activity and CNNM binding, did not [53,75]. Interestingly, the PRL-3 (C104D) mutant, which is deficient in phosphatase activity but capable of CNNM binding, still enhanced colonization, while the PRL-3 (R138E) mutant, which has phosphatase activity but is deficient in CNNM binding, did not [212]. These findings suggest that the oncogenic activity of PRLs may rely more on CNNM binding than on phosphatase activity, though further investigation is needed.

The exact mechanisms by which PRLs influence CNNMs remain unclear, and their role in magnesium transport is further complicated by emerging evidence of CNNM interactions with TRPM7, as discussed below. One proposed mechanism for PRL-mediated modulation of CNNM activity is altered protein localization. However, surface biotinylation experiments on HEK293 cells expressing CNNM3 with or without PRL-2 did not reveal differences in plasma membrane localization, suggesting that PRL binding may not alter CNNM trafficking or localization [155]. This does not exclude other local disruptions, such as inhibition of CNNM dimerization or interference with CNNM:TRPM7 interactions. Alternatively, PRLs may exert their effects by altering CNNM structure. The binding of PRL induces a twisted-to-flat conformation change in the CNNM CBS domain; a shift speculated to be transmitted through the protein to the transmembrane domain, similar to orthologous bacteria ion transporters. However, this hypothesis remains to be experimentally confirmed [45,211,213,214,215]. The flat orientation of the CBS domain orientation is thought to inhibit magnesium export, as mutations associated with hypomagnesemia in CNNM2 similarly lock the structure in a flat conformation, impairing magnesium extrusion [154,211,214].

While the precise mechanisms remain to be elucidated, it is increasingly evident that PRL proteins have functions beyond their phosphatase activity. Despite the high evolutionary conservation of their phosphatase domains, the primary purpose of PRL enzymatic activity is still debated. Phosphatases are challenging drug targets due to their shallow active sites, and PRLs are no exception. However, their minimal expression in adult tissue and upregulation in metastatic cancers make them attractive therapeutic targets. PRLs have been the focus of numerous drug discovery efforts, though most have met with limited success, often relying on phosphatase inhibition as a measure of efficacy [107,110,111,216,217,218,219]. Targeting the PRL–CNNM interaction offers a promising alternative. Assays are already being developed to identify inhibitors of this interaction [220], though it remains to be seen whether or not disrupting the PRL–CNNM interaction will be sufficient to negate the oncogenic effects of PRL proteins. Nevertheless, advances in understanding PRL biology over the past decade provide an optimistic outlook for therapeutic strategies targeting these proteins.

### 4.2. Competitive Binding and Divergent Regulation: ARL15 and CNNM Interactions

While investigating the PRL–CNNM interaction, a surprising parallel emerged: ARL15 also binds to CNNM proteins using its GTPase catalytic site. ARL15 binds to the CBS domain of CNNMs at a location adjacent to, and partially overlapping, the binding site of PRLs [221]. This CBS domain is critical for binding both PRLs and ARL15, potentially explaining its evolutionary significance. A key ARL15 point mutation (R95A) disrupts its interaction with CNNMs and is a valuable tool for studying this interaction in cell signaling. The structure of the ARL15–CBS domain complex has been deposited in the Protein Data Bank as 8F6D [53]. Interestingly, ARL15 binding to CNNM2 appears to inhibit the magnesium export properties of CNNM2, similar to PRL-3 [53]. However, as discussed later, the functional outcomes of PRLs and ARL15 binding CNNMs diverge in regulating TRPM7-mediated magnesium import. The details of the ARL15–CNNM interaction are still emerging and represent an active area of research.

Like PRLs, ARL15 can bind and co-immunoprecipitate all four CNNM family members in mixed overexpression systems. Endogenous CNNM3 has been shown to co-immunoprecipitate with endogenous ARL15 in HEK293T cells, confirming their interaction under basal conditions [221]. Recent proximity labeling assays across the ARF/ARL family further validated CNNM2–4 as ARL15 interacting partners in HEK and HeLa cell lines, highlighting the specificity of this interaction. Notably, no other ARF/ARL proteins labeled the endogenous CNNMs in this investigation, suggesting the exclusive role of ARL15 in regulating these proteins [222]. The crystal structure (8F6D) reveals that PRL and ARL15 binding sites on the CNNM proteins are adjacent and partially overlap (Figure 2), suggesting that these proteins compete for binding. In vitro isothermal titration calorimetry (ITC) analysis demonstrated that PRLs bind the CNNM CBS domain with nearly 100-fold affinity over ARL15, indicating PRLs likely outcompete ARL15 for CNNM binding. Additional ITC experiments and competition assays confirmed that ARL15 could not displace PRL-2 from the CBS domain [53].

Cell culture studies support the hypothesis that PRLs outcompete ARL15 for CNNM binding [33]. Hardy et al. (2023) showed that increasing PRL-3 expression in HEK293 cells reduced the amount of ARL15 co-immunoprecipitated with CNNM3. Interestingly, higher PRL-2 levels also correlated with decreased ARL15 expression, suggesting that ARL15 might be degraded when not bound to CNNMs. The stability of overexpressed ARL15 in HEK293 cells increased with proteasome inhibition and CNNM3 overexpression, indicating a potential CNNM-dependent stabilization of ARL15. These findings suggest that the PRLs not only outcompete ARL15 for CNNM binding but may also indirectly influence ARL15 stability through this competition.

The competition between PRLs and ARL15 for CNNM binding has significant functional consequences. The structure and signaling effects of the PRLs and ARL15 are compared and summarized in Figure 3. Despite binding overlapping sites on CNNMs, PRLs and ARL15 produce opposite effects on magnesium flux. Both inhibit CNNM-mediated magnesium export, but their impacts on magnesium uptake differ. PRLs enhance magnesium accumulation, while ARL15 reduces it. Zolotarov et al. (2021) demonstrated that ARL15 overexpression decreased radiolabeled magnesium uptake in kidney cancer cell lines, whereas ARL15 knockout increased uptake three-fold compared to controls [221]. These opposing effects of PRLs and ARL15 are further discussed in Section 4.3, which introduces the CNNM:TRPM7 interaction.

### 4.3. Structural and Functional Insights into CNNM and TRPM6/7 Complexes

The nature of the CNNM–TRPM7 interaction remains highly debated, with studies suggesting that CNNMs can function as activators [223], inhibitors [29], or neutral regulators [224] of TRPM7 activity. Although the precise structures of CNNM and TRPM6 or TRPM7 complexes are currently undetermined, multiple studies provide strong evidence of their interactions. Kollewe et al. (2021) used native PAGE to analyze TRPM7 complexes and observed a molecular weight of at least 1.2 MDa, exceeding the predicted size (850 kDa) of the TRPM7 tetramer [223]. Using cryo-slicing with quantitative mass spectrometry (csBN-MS), they identified complexes in rodent brains consisting of TRPM7, CNNM1-4, and ARL15 proteins, and, to a lesser extent, PRL-1/2 and TRPM6. Functional assays using a two-electrode voltage clamp (TEVC) in *Xenopus laevis* oocytes revealed that co-expression of ARL15 suppressed TRPM7 currents dose-dependently, while CNNM3 expression had no effect. In HEK293 cells, ARL15 overexpression significantly reduced TRPM7 activity in patch-clamp experiments, whereas CNNM3 overexpression did not alter TRPM7’s magnesium permeability. These findings suggest ARL15 inhibits TRPM7 activity, while CNNM3 is neutral in this context.

In parallel, Bai et al. (2021) conducted mass spectrometry in HEK293T cells and identified CNNMs, PRLs, and ARL15 as TRPM7-interacting partners [224]. Co-immunoprecipitation experiments confirmed the interaction between TRPM6/7 and all CNNM homologs. Endogenous CNNM2 and CNNM4 specifically co-immunoprecipitated with TRPM7, while CNNM3 and TRPM7 were shown to co-localize to the plasma membrane using immunofluorescence microscopy in HEK293T cells. Functional experiments with a fluorescent zinc reporter revealed that CNNM2 and CNNM4 enhanced TRPM7’s zinc uptake activity significantly more than CNNM1 or CNNM3. This effect required functional TRPM7 channels, as the inactive TRPM7 (E1047K) mutant did not respond to CNNM2 co-expression. Deletion of CCNM3 and CNNM4 in HEK293T cells also abolished TRPM7-mediated magnesium uptake, measured by magnesium-25, a stable magnesium isotope used for tracing, which was restored by the re-expression of CNNM4. Importantly, surface biotinylation confirmed that TRPM7 localization to the plasma membrane was unaffected by CNNM deletions, suggesting the loss of uptake was due to disrupted interactions between TRPM7 and CNNMs.

Interestingly, CNNMs also demonstrated TRPM7-independent roles in the above study. Overexpression of CNNMs reduced cellular magnesium levels independently of TRPM7 function, and the CNNM4 (S196P) mutant, associated with Jalili syndrome, impaired magnesium export while enhancing TRPM7 zinc influx [224]. These findings highlight CNNMs’ multifunctionality in magnesium regulation.

The effects of PRLs and ARL15 on TRPM7 activity were also explored using zinc influx assays [224]. PRLs enhanced TRPM7-mediated zinc uptake, while a PRL mutant deficient in CNNM binding, PRL-2 (R107E), failed to stimulate uptake. Knockout of CNNM3/4 significantly reduced PRL-2’s ability to enhance TRPM7 activity, confirming PRL reliance on CNNM interactions. Conversely, ARL15 suppressed TRPM7-mediated zinc uptake, with knockdown increasing activity. Although this study did not determine whether or not ARL15’s inhibition of TRPM7 is CNNM-dependent, the findings support a divergent regulation of TRPM7 by PRLs and ARL15.

Hardy et al. (2023) expanded on these findings, confirming that ARL15 enhances CNNM–TRPM7 interactions in HEK293 cells [33]. Co-immunoprecipitation assays showed that ARL15 overexpression increased CNNM pull-down with TRPM7. Additionally, TRPM7’s kinase activity was not required for these interactions, as wild-type and kinase-dead TRPM7 (K1648R) bound CNNMs and ARL15 equivalently. Interestingly, PRLs weakened CNNM–TRPM7 interactions. Using TRPM7-inducible HEK293 cells, the co-expression of CNNM3 with PRL-2 reduced CNNM3’s ability to bind TRPM7. The authors proposed that PRLs might facilitate CNNM trafficking from the plasma membrane to the cytosol, as observed through immunofluorescence microscopy. However, this hypothesis remains to be validated outside of overexpression systems, especially given conflicting findings that PRLs do not modulate CNNM localization [155].

Hardy et al. also developed a magnesium-sensitive biosensor based on the magnesium-responsive untranslated region (UTR) of PRL-2 mRNA, allowing real-time monitoring of cellular magnesium levels [33]. In this assay, high intracellular magnesium will inhibit the expression of a luciferase reporter. Inducible TRPM7 overexpression decreased biosensor luminescence, indicating increased intracellular magnesium. Co-overexpression of CNNMs partially restored luminescence, suggesting CNNMs may inhibit TRPM7-mediated magnesium influx. ARL15 alone had little effect on TRPM7 activity but enhanced CNNM3-mediated inhibition of TRPM7 when both proteins were co-expressed. Conversely, PRL-2 increased cellular magnesium and counteracted the CNNM-mediated inhibition of TRPM7 influx. Significantly, PRL-2 and CNNM3 mutants deficient in binding to one another—PRL-2 (D69A) and CNNM3 (D426A)—did not alter biosensor luminescence, demonstrating that their interactions were necessary for regulating intracellular magnesium levels and possibly TRPM7 activity.

While PRLs and ARL15 regulate TRPM7 in a CNNM-dependent manner, they exert opposite effects. PRLs can enhance TRPM7-mediated magnesium influx, while ARL15 inhibits it. However, the proposed mechanism of this effect is contested, with two competing models presented in the field. These models are described below.

### 4.4. Summary of Observations: Subunits vs. Localization Model

The above evidence supports two competing models to explain the PACT network’s function. Both models agree that PRLs and ARL15 differentially regulate TRPM7-mediated magnesium influx but diverge in their proposed mechanisms.

The first model (Figure 4A) proposes that CNNMs function as subunits of TRPM7 complexes, directly interacting with TRPM7 channels. In this model, PRLs and ARL15 bind to the CBS domain of CNNMs, inducing conformational changes that inhibit CNNM-mediated magnesium export. These structural changes are thought to be transmitted to the TRPM7 channel complex, modulating its activity. CNNMs thus act as signaling hubs within the TRPM7 complex, tuning channel activity while retaining their roles in magnesium export.

The second model (Figure 4B) hypothesizes that CNNMs and TRPM7 operate as independent proteins, with changes in CNNM localization mediating TRPM7 activity. ARL15 is proposed to enhance colocalization between CNNMs and TRPM7, thereby amplifying CNNMs’ inhibitory effect on magnesium influx. Conversely, PRLs outcompete ARL15 for CNNM binding in this model, reducing CNNM-mediated inhibition of TRPM7 and freeing the channel to promote magnesium influx. This model attributes TRPM7 regulation to spatial dynamics rather than conformational changes transmitted through a complex.

### 4.5. Alternative Interpretations and Conflicting Data

While the fine details of the PACT network continue to emerge, most investigations agree that PRLs enhance magnesium influx while ARL15 inhibits it, as described above. However, not all studies align with these conclusions, highlighting the need for further investigation.

Corre et al. (2018) reported findings that challenge the prevailing models. Using patch-clamp analysis, they observed that the overexpression of TRPM6 with ARL15 increased TRMP6 current activity compared to TRPM6 alone [225]. In contrast, overexpression of ARL15 (T46N), a GTPase-inactive mutant, did not produce the same effect. The authors noted that naïve HEK293T exhibited small baseline currents, and overexpression of ARL15 without TRPM6 did not alter current density. These results suggest that ARL15 might have an activating effect on TRPM6 activity.

Interestingly, the same study found that knockdown of the zebrafish ARL15 ortholog, *arl15b*, reduced total body magnesium content during early development. This phenotype was rescued by human ARL15 expression, indicating that ARL15 plays a role in magnesium homeostasis during development [225]. These findings are unexpected, as current models of the PACT network suggest ARL15 primarily inhibits CNNM-mediated efflux and TRPM7-mediated influx. Consequently, ARL15 knockdown would be predicted to increase magnesium content, making the observed decrease in *arl15b* knockdown zebrafish puzzling. This discrepancy underscores the complexity of the PACT network and suggests additional regulatory layers yet to be elucidated.

One aspect differentiating this study is the focus on TRPM6, which may have functional nuances distinct from TRPM7. Current models propose that ARL15 exerts its effects on TRPM6/7 via CNNMs. However, Bai et al. (2021) reported that while TRPM7 co-immunoprecipitated with all CNNM isoforms, TRPM6 only co-immunoprecipitated with CNNM1 and CNNM3 [154,224]. These data suggest TRPM6 may be less sensitive to CNNM-mediated, and thereby ARL15-mediated, regulation than TRPM7, though this remains speculative. This hypothesis does not fully explain ARL15’s observed activation of TRPM6 currents or the reduction in magnesium content in *arl15b* knockdown zebrafish. However, it highlights the need for future studies to include multiple homologs of the PACT proteins to clarify this complicated network.

The interactions of PACT proteins rely on their colocalization, with most activities seemingly occurring at the plasma membrane. However, in polarized epithelial cells of canonical absorption and transport tissues like the colon and renal systems, PACT protein localization may differ, raising questions regarding how these proteins might regulate each other’s activity. Polarized cells possess inherent directionality, with apical membranes facing the external environment and basolateral membranes interfacing with neighboring cells and the basal lamina [226]. Research on the differential localization of PACT proteins has primarily focused on CNNMs and TRPM6/7, given their roles in regulating ion flux. In polarized cells, CNNMs are predominantly localized to basolateral membranes, while TRPM7 is primarily apical [171]. For example, immunohistological evidence supports the basolateral localization of CNNM2 in distal convoluted tubule (DCT) cells [145,156,227]. Similarly, CNNM3 has been detected on the lateral membrane of proximal tubule cells in the kidneys of seawater pufferfish via immunohistochemistry (IHC) [228]. CNNM4 is predominantly localized to basolateral membranes in intestinal epithelial cells and ameloblasts, as shown by IHC and immunofluorescence analyses in mice [147,229]. In contrast, TRPM6 is apically localized in the DCT and the brush-border membrane of the small intestine [184], while TRPM7 is apically positioned on villus epithelial cells [230].

While CNNMs are generally described as basolateral and TRPM6/7 as apical, exceptions to this pattern exist. Transient overexpression of CNNM homologs in OK, Caco-2, and RPTEC/TERT1 cells showed varying degrees of apical localization, although CNNM4 remained predominantly basolateral [224]. Similarly, overexpressed murine TRPM7 exhibited a preference for basolateral localization [231]. These findings suggest CNNMs and TRPM7 may not be strictly separated in polarized epithelia, though this has not been confirmed in endogenous contexts. More research is needed to clarify the localization of the PACT proteins in polarized cells and determine the feasibility of the proposed model under physiological conditions.

## 5. Regulatory Mechanisms of the PACT Network

### 5.1. Magnesium Homeostasis Through Translational Regulation

The proteins involved in magnesium homeostasis, including PRLs, ARL15, CNNMs, and TRMP6/7, are dynamically regulated. Magnesium plays a significant regulatory role, particularly in translating PRLs and TRPM6/7 proteins. Multiple studies have shown that magnesium depletion increases PRL-2 expression levels in HeLa, HEK293, and HCT116 cell lines, an effect not observed with calcium depletion [209,232]. Similarly, a magnesium-deficient diet upregulates PRL-2 in the colon and the brush border of absorptive enterocytes of mice [233], suggesting a feedback mechanism that upregulates PRLs under magnesium-depleted conditions to maintain homeostasis.

The 5′ untranslated region (UTR) of the PRL mRNA is a key element in regulating PRL expression in a magnesium-sensitive manner [234]. Magnesium depletion significantly enhanced PRL-1/2 protein translation in HeLa cells despite no change in mRNA abundance. Metabolic labeling using a methionine analog L-azidohomoalanine confirmed increased PRL-1/2 protein translation under magnesium-depleted conditions. The magnesium-sensitive element within the 5′ UTR likely contains a small upstream open reading frame (uORF) responsible for ribosomal stalling under normal magnesium conditions. Interestingly, this uORF may encode for a nascent peptide critical for PRL translation, as rearranging its coding sequence disrupted magnesium sensitivity, even when the start and stop codons were preserved. Further investigation revealed that this magnesium-sensitive translation depends on the AMPK/mTORC2 pathways, as pharmacological or shRNA inhibition of AMPK abolished the enhanced PRL-1/2 translation under magnesium depletion.

In contrast, the regulation of ARL15 expression by magnesium level is poorly understood. Corre et al. (2018) reported that dietary magnesium levels did not affect ARL15 expression in mice, despite altering TRMP6 levels [225]. However, in zebrafish, *arl15b* showed tissue-specific responses to magnesium levels. Low magnesium diets slightly increased *arl15b* expression in the gills and brain, while high magnesium levels increased *arl15b* expression in gut tissues. This suggests that ARL15 may be a context- and tissue-specific manner of regulation.

CNNMs appear to be less dynamically regulated by magnesium than PRLs. Expression of CNNM3 and CNNM4 was unchanged in a panel of human cell lines exposed to varying magnesium concentrations [209,232]. However, CNNM2 expression increased in response to magnesium depletion in human leukemia cells [150] and zebrafish gut tissues [235].

Similar to PRLs, TRPM7 expression is regulated by magnesium through conserved magnesium-sensitive elements in upstream open reading frames (uORFs) [236]. These elements enhance TRPM7 expression in response to magnesium depletion. Nikonorova et al. (2014) proposed a mechanism in which the magnesium-sensitive uORF competes with the canonical protein-coding ORF, enabling the dynamic regulation of TRPM7 translation. TRPM6 is also upregulated in response to magnesium deficiency, as demonstrated in mice fed magnesium-deficient diets and zebrafish exposed to low-magnesium water [225,235]. These findings indicate that TRPM6/7 can be regulated at transcriptional and translational levels in response to magnesium availability.

Finally, Hardy et al. (2023) found evidence for mutual regulation among these proteins, further supporting their role as an interconnected network maintaining magnesium homeostasis [33]. Overexpression of any CNNM homolog or ARL15 in HEK293 cells increased endogenous PRL-1/2 expression, potentially due to reduced intracellular magnesium. Conversely, overexpression of TRPM7 decreased PRL-1/2 expression, possibly due to increased magnesium influx inhibiting PRL translation. These findings suggest a dynamic feedback network in which these proteins interact at multiple levels to regulate intracellular magnesium homeostasis.

### 5.2. Regulatory Roles of Enzymatic Activities

The phosphatase activity of the PRLs serves as an emerging regulator of the PRL–CNNM interaction. PRLs exhibit a unique “burst-kinetics” catalytic mechanism. Unlike most cysteine-based phosphatases, PRLs lack a serine or threonine residue in the enzyme’s P-loop, which is replaced by an alanine. This substitution slows their overall enzymatic activity, resulting in a prolonged release of the phospho-cysteine intermediate. While traditional cysteine-based phosphatases release the intermediate almost instantaneously, PRLs require over an hour to complete this step [70,152].

Importantly, PRLs cannot bind CNNMs in this phospho-cysteine intermediate state, suggesting that their enzymatic activity regulates this interaction [122,152,210]. Gulerez et al. (2016) [152] and Kozlov et al. (2020) [212] found that a significant percentage of endogenous PRLs exist in this intermediate state in both cell culture and mouse tissues. Under magnesium-deficient conditions, however, PRLs in HEK and HeLa cells lost their cystine phosphorylation, rendering them available for CNNM binding. Reintroducing magnesium rapidly drives PRLs back into the phospho-cysteine intermediate state, preventing further interaction with CNNMs. Since the enzymatic activity of PRLs is not directly sensitive to magnesium [122], the mechanism by which magnesium fluctuations influence the stability of the phospho-cysteine intermediate remains unclear.

ARL15, like PRLs, has enzymatic activity, though, in this case, through its GTPase function. However, this activity is similarly unconventional. ARL15 has a substitution in its active site—an alanine at position 86, where glutamine is typically found in other GTPases. This glutamine usually facilitates GTP hydrolysis, and its absence weakens ARL15’s intrinsic GTPase activity and reduces its affinity for GTP compared to canonical GTPases such as hRAS [140]. In contrast to the PRLs, ARL15’s enzymatic activity does not appear to inhibit its interaction with CNNM. Isothermal titration calorimetry (ITC) experiments demonstrated that ARL15 binding to CNNMs was unaffected in the presence of GTP [53]. While further studies are needed to confirm this observation within cellular contexts, the inhibition of ARL15–CNNM interactions through ARL15’s GTP binding seems unlikely. This property further distinguishes ARL15’s enzymatic role from that of the PRLs.

### 5.3. Regulation by Post-Translational Modifications: Lipidation and Glycosylation

Lipidation is a crucial post-translational modification (PTM) that can regulate protein localization within the cell, particularly to the plasma membrane [237]. PRLs and ARL15 undergo lipidation at their C- and N-termini, respectively [238,239], which is essential for their proper localization and function. The C-terminal CaaX motif of PRLs undergoes palmitoylation and prenylation (or farnesylation), directing these proteins to the plasma membrane. Experimental deletion of the CaaX motif causes PRLs to disperse into the cytosol and nucleus, resulting in a loss of enhanced cell migration and invasion [75,238]. Similarly, ARL15 is palmitoylated at several cysteine residues located at its N-terminus, primarily localizing the protein to the Golgi, with some distribution to the plasma membrane. Mutations in these cysteine residues lead to broad cytoplasmic distribution of ARL15 [239].

Although TRPM6 and TRPM7 are inherently membrane-associated due to their transmembrane domains, they also undergo palmitoylation near their TRP domains. Gao et al. (2022) found that preventing TRPM7 palmitoylation prevented its plasma membrane localization and ion flux capabilities [240], indicating the importance of this PTM for TRPM7 function.

Glycosylation is another important PTM that regulates interactions among these proteins, particularly within the CNNM family. N-glycosylation of extracellular domains is a common feature for trafficking transmembrane proteins, and CNNMs are no exception [241]. CNNM2 has a glycosylation site at N112 essential for plasma membrane localization; mutation of this site reduces its localization by 90% [227].

ARL15 may influence the glycosylation of CNNM proteins. Zolotarov et al. (2021) identified asparagine 73 (N73) as a key glycosylation site on CNNM3 and reported that the overexpression of ARL15 increased CNNM3’s complex glycosylation profile while decreasing its oligomannose glycoform. Lectin-gel shift assays and Western blots treated with linkage-specific glucosidase enzymes showed a molecular weight shift of CNNM3, suggesting glycosylation modifications. Overexpression of ARL15 in various kidney cancer cell lines increased CNNM3’s molecular weight, indicating enhanced glycosylation in the presence of ARL15 [231]. Conversely, CRISPR-mediated knockout of ARL15 decreased the amount of complex CNNM3 glycosylation without affecting total cellular glycosylation, suggesting a specific regulatory role.

Additionally, this complex glycosylation appears sensitive to magnesium concentrations. Magnesium depletion resulted in CNNM3 predominantly existing in its oligomannose form, while magnesium addition promoted complex glycosylation, further enhanced by ARL15 overexpression. Although these glycosylation changes did not seem to impact CNNM3’s localization to the plasma membrane, the authors suggest that such modifications might influence CNNM3’s magnesium flux capabilities. However, additional research is necessary to confirm ARL15’s role in modulating CNNM3 glycosylation and elucidate these modifications’ functional consequences. Future studies should employ quantitative methods to assess glycosylation changes and their impact on CNNM function. Investigating the molecular mechanisms underlying ARL15’s influence on CNNM glycosylation will also be crucial to determining its significance in magnesium homeostasis and its potential as a therapeutic target. A summary of reagulatory mechanisms that control the PACT network are shown in Figure 5. 

## 6. Conclusions and Future Perspectives

### 6.1. Consensus Aspects of the PACT Network

The PACT network represents a vital mechanism of magnesium homeostasis, with significant implications for development, cancer, and other diseases. Viewing this network as an integrated system rather than as isolated proteins offers valuable insights into its components’ regulatory functions and disease associations. Current evidence suggests several well-established aspects of the PACT network:CNNM Functions: CNNMs mediate magnesium export and import; each homolog exhibits distinct directionality preferences. Dimerization of CNNMs is necessary for magnesium export activity.PRL–CNNM Interaction: PRLs bind tightly to the CBS domains of CNNMs, inhibiting magnesium efflux. Overexpression of PRLs promotes magnesium influx via TRPM7 activation in a CNNM-dependent manner.ARL15–CNNM Interaction: ARL15 also binds to the CBS domains of CNNMs, inhibiting magnesium efflux. Unlike PRLs, ARL15 overexpression inhibits TRPM7 influx rather than activating it, again in a CNNM-dependent manner.Magnesium-Responsive Translation: The untranslated regions of PRLs and TRPM6/7 mRNAs contain magnesium-sensitive elements that enhance translation under low magnesium conditions, forming a feedback loop to maintain homeostasis.Functional Point Mutations: Mutations generated in PACT proteins have been invaluable for dissecting the interactions of this network. Table 1 summarizes these mutations, which target critical residues and have provided essential insights into the roles and regulatory mechanisms of the PACT proteins. These tools will continue to aid in the functional characterization of this complex system.

### 6.2. Open Questions and Future Directions

While much progress has been made, several aspects of the PACT network remain unresolved, necessitating further investigation. One of the central questions is how CNNMs mediate magnesium export. Structural studies are needed to clarify whether or not PRLs and ARL15 induce conformational shifts in CNNMs that regulate their magnesium transport activity, such as the transition between “open” and “closed” states. Additionally, the physiological roles of CNNMs and TRPM6/7 under normal conditions remain unclear. With minimal PRL expression in adult tissues, CNNMs, ARL15, and TRPM7 likely maintain magnesium homeostasis under basal conditions, while PRL overexpression in diseases such as cancer may disrupt this balance.

Another critical area of investigation is the nature of CNNM–TRPM7 interactions. Whether CNNMs function as subunits of TRPM7 or modulate the channel’s activity through colocalization remains a key question. Answering this will provide a deeper understanding of the regulatory mechanisms of the PACT network.

Accurate magnesium quantification remains a significant technical challenge. While ICP-MS provides highly sensitive measurements, its limitations highlight the importance of complementing it with other methods, such as fluorescent dyes and biosensors. These multi-method approaches will ensure more reliable data and facilitate progress in understanding magnesium dynamics.

Emerging evidence suggests that TRPM7’s influx activity extends to other divalent metals, raising questions about the broader implications of CNNM–TRPM7 interactions. Future studies should investigate whether or not CNNMs modulate the TRPM6/7-mediated flux of different metals and the potential biological significance of this cross-regulation.

Finally, the PACT network presents promising therapeutic opportunities. Current approaches primarily target individual proteins, such as PRLs and TRMP6/7, for therapeutic purposes. However, disrupting interactions within the network may offer alternative strategies for managing diseases linked to magnesium dysregulation, inducing cancer. As research advances, it will be crucial to move beyond broad generalizations of protein functions and consider the PACT network an interconnected system. This approach will reveal new insights into magnesium homeostasis and its implications for human health.

## Figures and Tables

**Figure 1 ijms-26-01528-f001:**
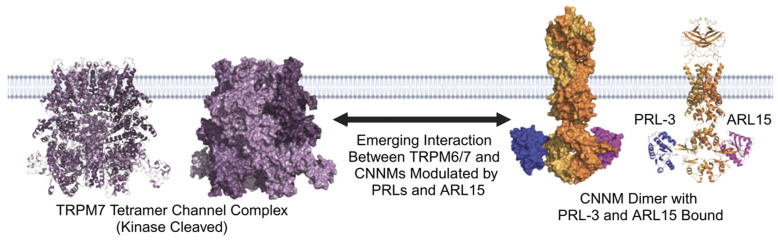
Molecular modeling of the PACT network. TRPM7 exists in a tetrameric complex with subunits shown in shades of purple, representing its kinase-cleaved form (based on PDB: 8SI2 [50]). The CNNM dimer, shown in shades of orange, is based on CNNM3 and was rendered using AlphaFold3 [51]. PRL-3 (blue) and ARL15 (magenta) bind to adjacent sites on the CBS domains of CNNMs (based on PDB: 5TSR [52] and 8F6D [53]). PRL and ARL15 interactions were overlaid on the predicted structure for illustrative purposes; it is important to note that these proteins compete for binding to CNNMs, and both are unlikely to bind simultaneously in physiological conditions.

**Figure 2 ijms-26-01528-f002:**
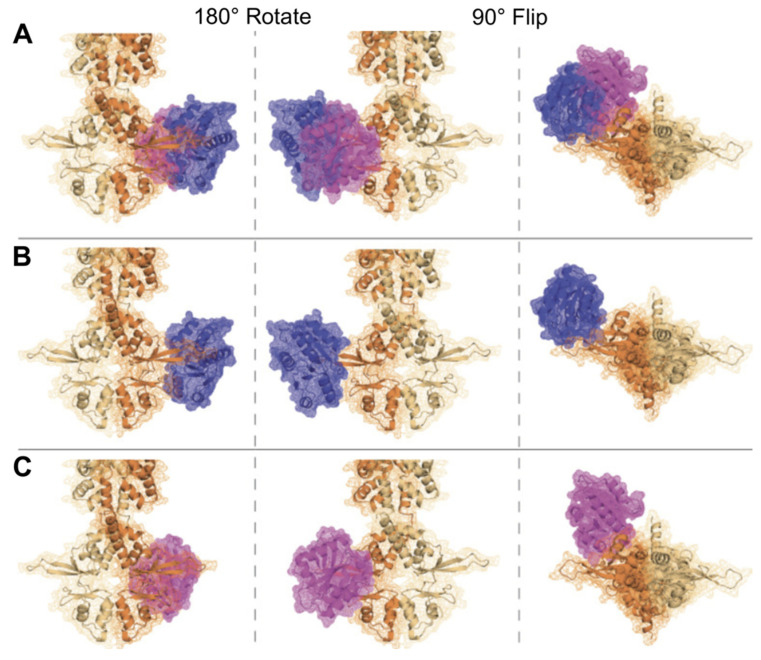
Molecular models of PRL-3 and ARL15 competing for CNNM binding at the CBS domain. The CNNM3 dimer (orange and tan) was rendered using Alphafold3 and adapted from PDB 5TSR and 8F6D. PRL-3 (blue) and ARL15 (magenta) were overlayed on the structure to compare binding locations and illustrate competition. (**A**) Both PRL-3 and ARL15 are shown overlaid on the CNNM3 dimer. (**B**) PRL-3 alone is overlaid. (**C**) ARL15 alone is overlaid.

**Figure 3 ijms-26-01528-f003:**
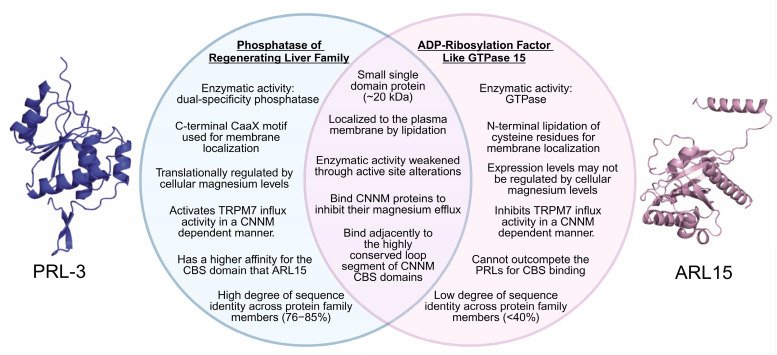
Comparative analysis of PRLs and ARL15. PRLs and ARL15 are comparable in size and share unique alterations that reduce their enzymatic activity. Both proteins inhibit CNNM-mediated magnesium export but exert opposite effects on TRPM7 activity, highlighting their divergent roles in regulating the PACT network.

**Figure 4 ijms-26-01528-f004:**
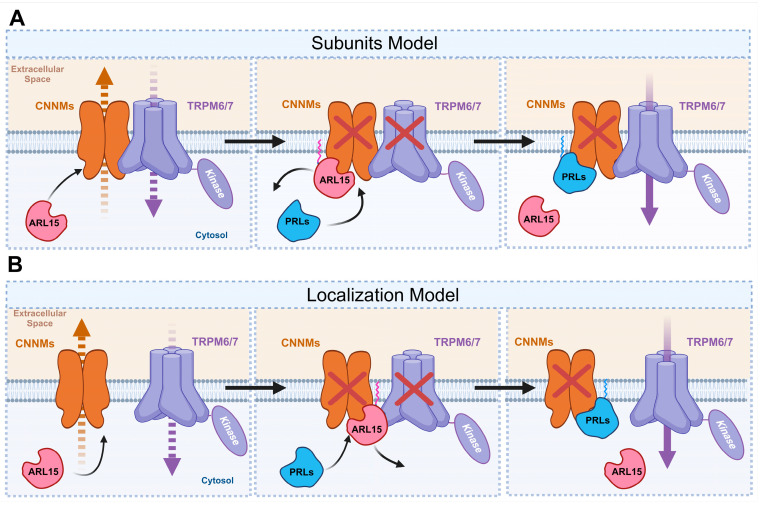
Proposed models of CNNM and TRPM7 interactions. (**A**) The subunit model suggests that CNNMs directly interact with TRPM7 as part of a complex. PRLs and ARL15 bind to the CBS domain of CNNMs, inducing conformational changes that inhibit CNNM magnesium export while modulating TRPM7 channel activity. (**B**) The localization model proposes that CNNMs inhibit TRPM7 activity primarily through co-localization rather than structural integration. ARL15 promotes CNNM–TRPM7 colocalization, reducing magnesium influx. PRL-3 competes with ARL15 for CNNM binding, freeing TRPM7 from CNNM-mediated inhibition and enhancing magnesium influx.

**Figure 5 ijms-26-01528-f005:**
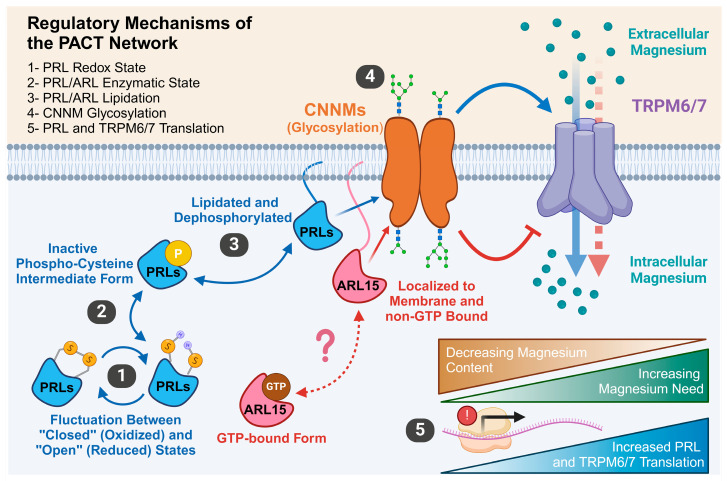
Summary of Regulatory Mechanisms Governing the PACT Network. (**1**) PRLs fluctuate between “open” (oxidized) and “closed” (reduced) redox states, which impacts substrate binding. (**2**) PRLs cannot bind to CNNMs in their enzymatic intermediate state; it is unclear if ARL15 shares this property. (**3**) Lipidation of PRLs and ARL15 localizes them to the plasma membrane, enabling their interaction with CNNMs. (**4**) Glycosylation events regulate CNNM functionality. (**5**) The translation rates of PRLs and TRPM6/7 are directly responsive to cellular magnesium concentration. Together, these mechanisms regulate the PACT network to help maintain cellular magnesium homeostasis.

**Table 1 ijms-26-01528-t001:** Key point mutations in PACT network proteins.

Protein	Residue and Mutation of Interest	Reported Significance	Reference
PRL-1	C49S	Forms a disulfide bond with C104 to inhibit phosphatase activity. Mutation to serine still retains phosphatase activity.	[152,242]
	E50R	Slight increase in phosphatase activity.	[121]
	C104S/D	Key catalytic cysteine residue (enzymatic nucleophile). The mutation of cysteine 104 to serine results in the loss of phosphatase activity. Mutation to aspartate inhibits phosphatase activity but does not inhibit CNNM binding.	[212,242]
	R110S	Key catalytic residue (phosphate coordination). Mutation to serine inhibits phosphatase activity.	[243]
	A111S	Enhances phosphatase activity	[243]
PRL-2	C46S/A	Cysteine 49 forms a disulfide bond with cysteine 101 to inhibit phosphatase activity. Mutation to serine can impair the ability to bind CNNM. Mutation to alanine does not inhibit CNNM binding.	[52,209]
	D69A	Key catalytic residue (general acid). Mutation to alanine can impair CNNM binding.	[114,209]
	C101S/A/D	Key catalytic cysteine residue (enzymatic nucleophile). Mutation to serine inhibits phosphatase activity and CNNM binding. Mutation to alanine inhibits enzymatic activity. Mutation to aspartate does not inhibit CNNM binding.	[114,152,209,212]
	R107E	Key catalytic residue (phosphate coordination). Mutation to glutamate can inhibit PRL-enhanced TRPM7 activity. Homologous mutation in PRL-3 inhibits CNNM binding.	[114,224]
	A111S + V113S	Enhances phosphatase activity. A111S + V113S double mutation increases phosphatase activity 24-fold.	[243]
	C170S + C171S	Prevents farnesylation and causes cytosolic distribution. The C170S and C171S double mutant encourages active site reduction to enhance enzymatic activity in vitro.	[244,245]
PRL-3	C49A/S	Readily forms a disulfide bond with C104 to inhibit phosphatase activity. Both weaken PRL affinity for CNNM, with serine having the worst affinity.	[111,153,246]
	E50R	Enhances enzymatic activity but also slightly destabilizes the protein.	[121]
	Y53F	Potential site of SRC-mediated phosphorylation. Y53F impairs oncogenic functions.	[247]
	D72A	Key catalytic residue (general acid). Mutation to alanine causes loss of phosphatase activity.	[246]
	C104A/S/E/D	Key catalytic and CNNM binding residue. Mutations to serine and glutamate inhibit CNNM binding, with glutamate having the lowest affinity. Mutation to alanine has a weaker affinity than wild type but can still bind in vitro and in crystal structures. Mutation to aspartate inhibits enzymatic activity but can still allow CNNM binding. Mutation to aspartate or glutamate is not detrimental to trafficking or thermal stability.	[52,153,212,248]
	A106V	Weakens phosphatase activity.	[111]
	R110A/E	Key catalytic residue (phosphate coordination). Mutation to alanine combined with E50R leads to loss of phosphatase activity towards synthetic substrates while retaining wildtype activity against PIP2. Mutation to glutamate inhibits both phosphatase activity and CNNM binding.	[71,121]
	A111S	Enhanced phosphatase activity but may inhibit activity against PIPs. Wild-type affinity for CNNM.	[52,120,246]
	R138E	Normal phosphatase activity but inhibits CNNM binding. Mutation to glutamate does not negatively affect trafficking or thermal stability.	[52,212]
	C170S + C171S	Prevents farnesylation and causes cytosolic distribution. C170S appears to be the primary site of lipidation. Deletion of C170-M173 enhances phosphatase activity in vitro.	[75,244,249]
ARL15	C22Y + C23Y	Palmitoylation sites for Golgi localization. Mutation to Y prevents modification.	[239]
	T46N	Prevents release of GDP, locked in “inactive” form.	[140]
	A86L	Inhibits GTP hydrolysis, locked in “active” form.	[140]
	S94K/W	Decreases affinity for CBS domain by 20-fold. Does not Co-IP with CBS domains of CNNM2/3/4.	[53]
	R95A	Mutation prevents CNNM binding (ITC and Co-IP). Cannot inhibit CNNM2-mediated magnesium efflux and cannot inhibit TRPM7-mediated zinc influx. Does not disrupt plasma membrane localization.	[53]
	P130W	Mutation prevents CNNM binding (ITC and Co-IP).	[53]
CNNM1	C508F	CNNM2/3/4 have P at this position. Mutation of C508 to F enhances ARL15 binding, as is evident through Co-IP, and there is a 3-fold increase in affinity measured by ITC.	[53]
CNNM2	L48P	Extracellular domain mutation. Inhibits CNNM2-mediated magnesium uptake. Hampers plasma membrane localization.	[162]
	N112A	Glycosylation site that stabilizes the protein at the plasma membrane. Mutations prevent glycosylation and membrane expression.	[227]
	E122K	Pathogenic mutation (hypomagnesemia), impairs magnesium uptake and reduces membrane expression.	[157]
	S269W	Pathogenic mutation (hypomagnesemia), impairs magnesium uptake.	[157]
	V324M	Transmembrane domain mutation. Inhibits CNNM2-mediated magnesium uptake. Hampers plasma membrane localization.	[162]
	L330F	Pathogenic mutation (hypomagnesemia), retains magnesium uptake capabilities.	[157]
	G356A	Magnesium binding site mutation that does not hamper expression or localization. Inhibits magnesium export activity and does not stimulate TRPM7 activity.	[224,250]
	E357A/K	Magnesium binding site mutation that does not hamper expression or localization. Inhibits magnesium export activity and does not stimulate TRPM7 activity. Mutation to K is a pathogenic mutation (hypomagnesemia) and impairs magnesium uptake.	[157,224,250]
	P360A	Magnesium binding site mutation that does not hamper expression or localization. Inhibits magnesium export activity.	[250]
	L418P	Acidic helical bundle mutation. Inhibits CNNM2-mediated magnesium uptake. Hampers plasma membrane localization.	[162]
	H523K	Mutation that can prevent ARL15 binding as measured by Co-IP and ITC. Retains near wild-type affinity for PRL-2.	[53]
	F524K	Mutation that can prevent ARL15 binding as measured by Co-IP and ITC.	[53]
	D558A	Binds to ARL15 with WT affinity while preventing PRL binding (ITC).	[53]
	T568I	Pathogenic mutation (hypomagnesemia) found in the CBS domain inhibits magnesium uptake and export. Does not affect plasma membrane localization. Cannot stimulate TRPM7 influx. Proposed to lock the CBS domain in a “flat” conformation. PRL-2 can still bind to this mutant, as determined by ITC. This mutation does not impair the ability of PRL to outcompete ARL15 for binding (NMR).	[53,154,156,215,224]
	S795L	CNBH domain mutation. Inhibits CNNM2-mediated magnesium uptake. Hampers plasma membrane localization.	[162]
CNNM3	N73A	Glycosylation site. Mutation to alanine decreases co-localization with ARL15 and has reduced localization to the plasma membrane.	[221]
	H391K	Mutation that can prevent ARL15 binding, as measured by Co-IP and ITC.	[53]
	F392K	Mutation that can prevent ARL15 binding, as measured by Co-IP and ITC.	[53]
	D426A	Inhibits PRL binding. Does not inhibit PRL phosphatase activity in vitro. Overexpression impairs the ability of cells to grow in magnesium-depleted media. WT CNNM3 overexpression can enhance in vivo tumor size, but D426A decreases tumor size compared to the control (DB-7 model).	[152,155]
	P427A	Weakens PRL affinity. Partially inhibits PRL phosphatase activity in vitro.	[152]
	G433D	Mutation in CBS domain that can prevent PRL interaction. Reduces magnesium influx.	[209]
	L575K	CNBH mutation, can inhibit dimerization	[251]
	F577K	CNBH mutation, can inhibit dimerization	[251]
CNNM4	S196P	Pathogenic mutation (Jalili syndrome). Magnesium binding site mutation that does not hamper expression or localization. Inhibits magnesium export activity but seemingly enhances TRPM7 activity over WT CNNM4.	[167,224,250]
	S200Y	Pathogenic mutation in the DUF21 domain (Jalili syndrome), diminishes Mg efflux capabilities. Magnesium binding site mutation that does not hamper expression or localization. Inhibits magnesium export activity and does not stimulate TRPM7 activity.	[147,224,250]
	N250A	Magnesium binding site mutation that does not hamper expression or localization. Inhibits magnesium export activity and does not stimulate TRPM7 activity.	[224,250]
	N254A	Potential sodium binding site. Weakens but does not abolish magnesium export activity.	[250]
	L324P	Pathogenic mutation in the DUF21 domain (Jalili syndrome), diminishes Mg efflux capabilities.	[147]
	R407L	Disease-associated mutation (Jalili syndrome). Mutation in Mg-ATP binding site prevents ATP binding. Abolishes magnesium efflux ability. Unable to dimerize.	[144,252]
	H450K	Mutation that can prevent ARL15 binding, as measured by Co-IP and ITC.	[53]
	F451K	Mutation that can prevent ARL15 binding, as measured by Co-IP and ITC.	[53]
	D485A	Inhibits PRL interaction. Export activity cannot be blocked by PRLs.	[152]
	P486A	Similar effect as F487A. Weakens PRL interaction. PRL can partially block export activity.	[152]
	F487A	Similar effect as P486A. Weakens PRL interaction. PRL can partially block export activity.	[152]
	T495I	Equivalent to T569I in CNNM4, disease-associated mutation. Inhibits magnesium efflux. Does not affect plasma membrane localization. Mutation in Mg-ATP binding site prevents ATP binding. Prevents dimerization.	[144,154]
	F557K	CNBH domain mutation. Reduces but does not abolish magnesium efflux activity.	[144]
	F631K	CNBH mutation, can inhibit dimerization, inhibits magnesium efflux, does not affect plasma membrane localization. Does not enhance TRPM7 activity.	[144,251]
	M629K	CNBH mutation, can inhibit dimerization, does not inhibit magnesium efflux, does not affect plasma membrane localization.	[251]
TRPM6	S141L	Pathogenic mutation (hypomagnesemia), loss of channel activity, prevents oligomerization of the channel.	[179]
	P1017R	Pathogenic mutation in the pore (hypomagnesemia), suppresses TRPM7 activity, does not affect channel assembly or trafficking.	[253]
	E1024Q	Pore mutation, inhibits calcium and magnesium permeability.	[254]
	E1029Q	Pore mutation, inhibits calcium and magnesium permeability.	[254]
	K1085Q	Diminished channel activity, possibly due to loss of PIP binding.	[255]
	R1088Q	Diminished channel activity, possibly due to loss of PIP binding.	[255]
	K1098Q	Complete loss of channel function, possibly due to loss of PIP binding.	[255]
	K1804R	Homolog to the K1648R mutation in TRPM7. Does not seem to be phosphorylated by TRPM7.	[256]
TRPM7	S138L	Inhibits channel oligomerization.	[179]
	A931T	Pathogenic mutation (Trigeminal neuralgia). Results in hyperexcitability as well as sustained sodium influx. May destabilize a hydrophobic ring near the proteins’ voltage-sensing domain.	[196]
	P1040R	Functions as a dominant negative to inhibit channel activity and does not affect channel assembly or trafficking.	[253]
	E1047Q	Pore mutation, eliminates calcium and magnesium permeability.	[254]
	E1052Q	Pore mutation, decreases calcium and magnesium permeability.	[254]
	S1107E/R/K/Q	Constitutively active channel, insensitive to inhibition by high levels of magnesium or pH fluctuations. E/R mutations are less sensitive to PIP2 depletion compared to WT. May strengthen channel/PIP interactions.	[257]
	N1097Q (mouse)	(Mouse) Gain of function mutation. Channel is constitutively active and has high activity. Mutation increases the frequency of channel opening. Still subject to pharmacological inhibition.	[50,258]
	K1112Q	Reduced channel activity, possibly due to disruption of PIP binding.	[255]
	R1115Q	Normal channel activity, possibly disrupts PIP interactions.	[255]
	C1143,1144,1146	Sites of palmitoylation. Mutating all three to alanine prevents plasma membrane localization.	[240]
	S1208D	Potential site of phosphorylation. Mutation to aspartate enhances channel current.	[223]
	K1225Q	Normal channel activity, possibly disrupts PIP interactions.	[255]
	T1482I	Pathogenic mutation (Guamanian amyotrophic lateral sclerosis and parkinsonism dementia). Normal kinase activity, generates normal channels that have a heightened sensitivity of inhibition by magnesium.	[259]
	S1496D	Potential site of phosphorylation. Mutation to aspartate enhances channel current.	[223]
	D1510A	Caspase cleavage point to release kinase domain. Mutation prevents cleavage.	[260]
	S1565D	Autophosphorylation site. Phosphomimetic mutation. Disrupts kinase activity, does not disrupt kinase domain dimerization. S1565A does not have the same effects. Possible regulatory switch controlling kinase activity.	[178]
	S1567D	Potential site of phosphorylation. Mutation to aspartate enhances channel current.	[223]
	K1648R	Kinase-deficient mutant, does not impair channel activity, deficient in magnesium-dependent suppression of the channel (equivalent to mouse K1646R).	[180]
	S1777D	Autophosphorylation site. Phosphomimetic mutation. Disrupts kinase activity, S1777A increases kinase activity.	[178]
	G1779D	Kinase-deficient mutant, does not impair channel activity, deficient in magnesium-dependent suppression of the channel.	[261]

## Data Availability

All data presented in this review are available in the manuscript.

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
