# Peer review of "The PACT Network: PRL, ARL, CNNM, and TRPM Proteins in Magnesium Transport and Disease"

_ijms, 2025, doi:10.3390/ijms26041528_

Round 1

Reviewer 1 Report

Comments and Suggestions for Authors

This is a well-written and documented review of the current knowledge on PACT proteins involved in magnesium homeostasis. The figures are clear and helpful for readers' comprehension, and reference list covers almost all information with some important gaps (see below) that should be corrected.

Intriguingly, and despite an extensive description of the TRPM/CNNM interaction is included in the text, no mention about the different CNNM and TRPM7 localization in epithelial cells (basolateral vs apical) is mentioned or commented in this work. This omission, which is also systematically omitted in the literature, is relevant as it highlights contradictions in the current models proposing a direct interaction between these two proteins in particular cell types. Authors' comments on this fact will be well received (including the corresponding references).

Major concerns:

As shown in the present form, the authors' comments on CNNM4 might make readers infer that the inhibition or knockdown of this gene always results in oncogenic promotion, which is not true. Several recent findings indicate the opposite effect when the inhibition is done tissue-specific. See for example, the studies by Simón J, et al, J Hepatol. 2021 Jul;75(1):34-45; González-Recio et alNat Commun. 2022 Nov 25;13(1):6816; or González-Recio et alHepatology. 2024 Nov 19. doi: 10.1097/HEP.0000000000001156.). All this information should be included in the revised version of the manuscript.

- Page 12. section 4.4: 4.4. Competing Models: Subunits vs. Localization: authors summarize two competing models on the CNNMs-TRPM7 interaction role. However, no references supporting each model are indicated along this section, which should be included.

Minor points: 

- Please revise and correct some typos, for example in page 8: "These findings initially supported the hypothesis that PRLs i increase intracellular magnesium by inhibiting CNNM-mediated export.

Another one also in page 8: "...suggesting a PR-dependent role in magnesium up-take"...

Author Response

We thank Reviewer 1 for their positive feedback and constructive suggestions, which have helped to strengthen the manuscript. Below, we describe the changes made in response to each comment, with the corresponding revisions highlighted in the text for clarity.  

Q1: Intriguingly, and despite an extensive description of the TRPM/CNNM interaction is included in the text, no mention about the different CNNM and TRPM7 localization in epithelial cells (basolateral vs apical) is mentioned or commented in this work. This omission, which is also systematically omitted in the literature, is relevant as it highlights contradictions in the current models proposing a direct interaction between these two proteins in particular cell types. Authors' comments on this fact will be well received (including the corresponding references).

Author Response: We appreciate the reviewer’s insightful comment and agree that this discussion enhances the manuscript by addressing a critical gap in the current understanding of CNNM and TRPM7 interactions. We have incorporated this information into the "Alternative Interpretations and Conflicting Data" section ot highlight how the differential localization of CNNMs and TRPM7 in polarized epithelial cells presents a potential limitation to the proposed models. Additionally, we have included evidence that challenges this generalization, emphasizing the complexity and the need for further research in this area, which aligns with our manuscripts ultimate objective. The corresponding revisions can be found on lines 624-648. 

Q2:  As shown in the present form, the authors' comments on CNNM4 might make readers infer that the inhibition or knockdown of this gene always results in oncogenic promotion, which is not true. Several recent findings indicate the opposite effect when the inhibition is done tissue-specific. See for example, the studies by Simón J, et al, J Hepatol. 2021 Jul;75(1):34-45; González-Recio et al , Nat Commun. 2022 Nov 25;13(1):6816; or González-Recio et al, Hepatology. 2024 Nov 19. doi: 10.1097/HEP.0000000000001156.). All this information should be included in the revised version of the manuscript.

Response: We included additional information on line 255 referencing the 2022+2024 manuscripts by González-Recio et al as evidence that CNNM4 reduction does not necessarily result in oncogenesis. Furthermore, we found that Simón J, et al, J Hepatol. 2021 Jul;75(1):34-45 provided an additional reference for the disruption of CNNM4 resulting in the dysregulation of magnesium homeostasis and included this reference in line 260. We thank the reviewer for these additions. 

Q3: Page 12. section 4.4: 4.4. Competing Models: Subunits vs. Localization: authors summarize two competing models on the CNNMs-TRPM7 interaction role. However, no references supporting each model are indicated along this section, which should be included.

Response: Thank you for your suggestion. In section 4.4, we aim to summarize the key observations presented throughout “4. Molecular Basis of the PACT Network,” into a concise and digestible overview. These insights are drawn from ~50 references cited in part 4. To clarify this intent, we have updated the subheading of 4.4 to “Summary of Observations: Subunits vs Localization Model”, line 551. This change emphasizes that the section serves as a synthesis of the referenced data rather than introducing new citations. 

Q4: Please revise and correct some typos, for example in page 8: "These findings initially supported the hypothesis that PRLs i increase intracellular magnesium by inhibiting CNNM-mediated export. Another one also in page 8: "...suggesting a PR-dependent role in magnesium up-take"...

Response: We thank the reviewer for their attention to detail and have reviewed the document with to ensure no additional typos remain in the text. 

Reviewer 2 Report

Comments and Suggestions for Authors

The paper by Jolly and co-authors provides a reasonable overview of magnesium ion transport and homeostasis systems in animal cells.

Major Comments:

1.      The systems described are quite complex, so the review would have benefited if the authors had described the evolution of this complexity, i.e. which of the systems described have homologous precursors in bacteria or archaea. In this way, it would have been possible to assess which of the systems are basic and which are the result of further evolution of complexity in eukaryotes or within metazoans.

2.      It might be useful to show as many protein structures as possible, especially as this is not a major problem with the public release of AlphaFold UniProt.

3.      The descriptions of regulatory mechanisms should be complemented with schemes in which arrows show positive and negative interactions between the components.

Minor comment:

4.      Lines 73-74: The statement "Approximately 80% of cytosolic magnesium is bound to ATP, forming a biologically active complex' is not referenced. The reference should be provided and, if possible, it should be explained how this result was obtained.

Lines 90-96: At the very first presentation of transport systems, their rather weird names are best taken in quotes, e.g.: ‘mitochondrial RNA splicing 2’.

Author Response

We thank Reviewer 2 for their positive feedback and constructive suggestions, which have helped to strengthen the manuscript. Below, we describe the changes made in response to each comment, with the corresponding revisions highlighted in the text for clarity. 

Q1: The systems described are quite complex, so the review would have benefited if the authors had described the evolution of this complexity, i.e. which of the systems described have homologous precursors in bacteria or archaea. In this way, it would have been possible to assess which of the systems are basic and which are the result of further evolution of complexity in eukaryotes or within metazoans.

Response: We appreciate the reviewer’s thoughts on this. The lack of PRL and ARL15 homologs in bacteria, while CNNM homologs are present, is interesting. Since the evolutionary trajectory of the magnesium transporters has been expertly reviewed elsewhere https://doi.org/10.1007/s00018-022-04442-8 , we added some text with additional references to briefly describe the conservation of these proteins, lines 224-229. 

Q2: It might be useful to show as many protein structures as possible, especially as this is not a major problem with the public release of AlphaFold UniProt.

Response: We agree with the reviewer that the presentation of protein structures is an excellent way to communicate the nature of this network. We have presented either AlphaFold models or adapted PDB structures of at least 1 homolog from each member of the PACT network, with some being presented multiple times. The structural depiction of all the family members, such as all three PRLs, would be somewhat redundant due to their high degree of similarity. We present several protein:protein interactions as adaptations of the previously published structures of the PRL:CNNM interaction, ARL15:CNNM interaction, a model of the proposed CNNM dimer as well as the published heterometric TRPM7 channel (kinase cleaved). Unfortunately, interactions between CNNM:TRPM6/7 have not been confirmed by a published structure, and the very large sizes of the heterocomplexes (>10,000 amino acids) is beyond the scope of what the public release of AlphaFold can model (~5000 amino acids). 

Q3: The descriptions of regulatory mechanisms should be complemented with schemes in which arrows show positive and negative interactions between the components.

Author Response: Thank you for this valuable suggestion. We agree that visualizing the regulatory mechanisms with a schematic enhances the clarity of the manuscript. To address this, we have included Figure 5, which provides a summary of the discussed regulatory mechanisms. (Lines 788-795). 

Q4: Lines 73-74: The statement "Approximately 80% of cytosolic magnesium is bound to ATP, forming a biologically active complex' is not referenced. The reference should be provided and, if possible, it should be explained how this result was obtained.

Response: We have updated this section with references and provided more details regarding the freely available (ionized) concentrations, lines 70-73.

Q5: Lines 90-96: At the very first presentation of transport systems, their rather weird names are best taken in quotes, e.g.: 'mitochondrial RNA splicing 2'.

Response: We have made these changes, lines 92-100. 

Round 2

Reviewer 1 Report

Comments and Suggestions for Authors

In Figure 1, the authors present a ribbon representation on the right side depicting the binding modes of PRL3 and ARL15 proteins, indicating they compete for the same binding site on the CNNM transporter. However, as currently illustrated, the figure is incorrect; it shows identical binding modes for PRL and ARL15, and this is not correct (the orientation of the alpha-helices of ARL15 does not coincide with the crystal structure of the CNNM-ARL15 complex. Thus, as currently represented, it seems that the authors have placed two PRL subunits bound to two Bateman modules of the CNNM dimer. This error needs to be corrected, as it is misleading. Moreover, this Figure 1 panel  is inconsistent with the ARL15/PRL binding modes presented in panels B and C of Figure 2.

In Section 4.2, there is once again a misinterpretation of the referenced publication: the authors attribute to Zolotarov et al. the claim that "ARL15 binds to CNNM proteins using its catalytic GTPase site, through the same loop of the Bateman domains of CNNMs that interacts with PRLs." While part of this loop does maintain some interactions with ARL15 in the CNNM2 structure modeled by Zolotarov et al., the statement, as written, differs significantly from what is mentioned in that paper. To the best of my knowledge, no published article has claimed that the binding mode of ARL15 and PRL1 is identical and occurs through the same amino acids of the loop of CBS2 domain, as erroneously shown on the right side of Fig 1. Please review this.

Author Response

Q1: In Figure 1, the authors present a ribbon representation on the right side depicting the binding modes of PRL3 and ARL15 proteins, indicating they compete for the same binding site on the CNNM transporter. However, as currently illustrated, the figure is incorrect; it shows identical binding modes for PRL and ARL15, and this is not correct (the orientation of the alpha-helices of ARL15 does not coincide with the crystal structure of the CNNM-ARL15 complex. Thus, as currently represented, it seems that the authors have placed two PRL subunits bound to two Bateman modules of the CNNM dimer. This error needs to be corrected, as it is misleading. Moreover, this Figure 1 panel  is inconsistent with the ARL15/PRL binding modes presented in panels B and C of Figure 2.

Author Response: We thank the reviewer for their careful attention to detail. We have updated Figure #1 accordingly. (Line 413). 

Q2: In Section 4.2, there is once again a misinterpretation of the referenced publication: the authors attribute to Zolotarov et al. the claim that "ARL15 binds to CNNM proteins using its catalytic GTPase site, through the same loop of the Bateman domains of CNNMs that interacts with PRLs." While part of this loop does maintain some interactions with ARL15 in the CNNM2 structure modeled by Zolotarov et al., the statement, as written, differs significantly from what is mentioned in that paper. To the best of my knowledge, no published article has claimed that the binding mode of ARL15 and PRL1 is identical and occurs through the same amino acids of the loop of CBS2 domain, as erroneously shown on the right side of Fig 1. Please review this.

Author Response: We have updated the text to better reflect the referenced manuscript regarding the nature of the ARL15:CNNM interaction. (Line 421). We also updated the text in Figure 3 to better communicate that the PRL:ARL binding sites on CNNMs are adjacent and not identical binding modes. (Line 467)